

# Evolution of protein domain repertoires of CALHM6

Aneela Javed,  Sabahat Habib* and  Aaima Ayub*

Molecular Immunology Laboratory, Department of Healthcare Biotechnology, Atta-ur-Rahman School of Applied Biosciences (ASAB), National University of Sciences and Technology (NUST), Islamabad, Pakistan
* These authors contributed equally to this work.

## ABSTRACT

Calcium ($Ca^{2+}$) homeostasis is essential in conducting various cellular processes including nerve transmission, muscular movement, and immune response. Changes in $Ca^{2+}$ concentration in the cytoplasm are significant in bringing about various immune responses such as pathogen clearance and apoptosis. Various key players are involved in calcium homeostasis such as calcium binders, pumps, and channels. Sequence-based evolutionary information has recently been exploited to predict the biophysical behaviors of proteins, giving critical clues about their functionality. Ion channels are reportedly the first channels developed during evolution. Calcium homeostasis modulator protein 6 (CALHM6) is one such channel. Comprised of a single domain called Ca_hom_mod, CALHM6 is a stable protein interacting with various other proteins in calcium regulation. No previous attempt has been made to trace the exact evolutionary events in the domain of CALHM6, leaving plenty of room for exploring its evolution across a wide range of organisms. The current study aims to answer the questions by employing a computational-based strategy that used profile Hidden Markov Models (HMMs) to scan for the CALHM6 domain, integrated the data with a time-calibrated phylogenetic tree using BEAST and Mesquite, and visualized through iTOL. Around 4,000 domains were identified, and 14,000 domain gain, loss, and duplication events were observed at the end which also included various protein domains other than CALHM6. The data were analyzed concerning CALHM6 evolution as well as the domain gain, loss, and duplication of its interacting partners: Calpain, Vinculin, protein S100-A7, Thioredoxin, Peroxiredoxin, and Calmodulin-like protein 5. Duplication events of CALHM6 near higher eukaryotes showed its increasing complexity in structure and function. This *in-silico* phylogenetic approach applied to trace the evolution of CALHM6 was an effective approach to get a better understanding of the protein CALHM6.

Corresponding author
Aneela Javed,
javedaneela19@gmail.com

## INTRODUCTION

Calcium ($Ca^{2+}$) homeostasis is an established mechanism important in conducting various processes *e.g.*, nerve transmission in the nervous system, contraction, and relaxation in cardiac, smooth, and skeletal muscles, exocytosis, apoptosis, transcription, and in the

immune system. Even subtle changes in $Ca^{2+}$ concentration in the cytoplasm significantly affects responses elicited by the immune system such as antibody secretion by lymphocytes, bacterial clearance by neutrophils, and apoptosis (*Feher, 2017*; *Grinstein & Klip, 1989*; *Puzianowska-Kuznicka & Kuznicki, 2009*). Hence a strict regulatory system is maintained for $Ca^{2+}$ concentration through calcium channels, buffers, pumps, binding proteins, and exchangers between intracellular $Ca^{2+}$ stores, plasma membrane, and cytoplasm (*Puzianowska-Kuznicka & Kuznicki, 2009*). Calcium pumps include sarco-endoplasmic reticulum $Ca^{2+}$-ATPases (SERCA) while calcium-binding proteins include Calmodulin, *etc* (*Puzianowska-Kuznicka & Kuznicki, 2009*; *Schnellmann & Covington, 2010*). Two types of $Ca^{2+}$ channels are active: Voltage-gated and receptor-activated calcium channels (RACCs) and are further branched into several classes. There are different types of RACCs such as ryanodine receptors (RyRs) and inositol-1,4,5-triphosphate (IP3) receptors (IP3Rs) which release calcium, while ten members of the voltage-operated calcium channels (VOCCs) each serve a discrete role in physiological events (Fig. 1) (*Puzianowska-Kuznicka & Kuznicki, 2009*; *Catterall, 2011*).

All the proteins involved in calcium homeostasis work together by their similar protein and domain structures. Understanding the evolutionary past of these structures can give a clear insight into the functional importance of various domains and proteins (*Bordin et al., 2021*). Studies have reported the possibility of the very first ion channels developed by prokaryotes to be $Ca^{2+}$ ion channels, an argument further supported by the phylogenetic age of these channels and their presence in many prokaryotes compared to others. The bacteria belonging to the genus Rickettsia entered an endosymbiotic relationship with early eukaryotes, now known as mitochondria, and began working as calcium-regulating organelles to develop more pathways. Gradually more $Ca^{2+}$ regulatory systems began developing to maintain primitive programmed cell death, exocytosis, homeostasis, and eventually more complex functions, each requiring different types of protein systems to be developed over time (*Case et al., 2007*). One such calcium channel protein is CALHM6 identified in 2010, which is involved in the diverse immune modulatory function.

CALHM6 or family with sequence similarity 26, member F (FAM26F) belongs to a gene family of calcium homeostasis modulators (CALHM) also known as the FAM26 gene family (*Ebihara et al., 2010*). It is comprised of a total of six genes sharing 20 to 50 percent sequence similarity among themselves but lacks similarity to other genes, localized as two clusters on two different chromosomes (*Grinstein & Klip, 1989*; *Puzianowska-Kuznicka & Kuznicki, 2009*). These six genes are: CALHM3 (FAM26A), CALHM2 (FAM26B), and CALHM1 (FAM26C) present on chromosome 10, whereas CALHM4 (FAM26D), CALHM5 (FAM26E), and CALHM6 (FAM26F), the focus of the study, are present on chromosome 6 (Table 1) (*Grinstein & Klip, 1989*; *Schnellmann & Covington, 2010*). CALHM6 is 315 amino acids long protein with a molecular weight of 34.258 kD and consists of an immunoglobulin-like fold and 3–5 transmembrane helices (*Ebihara et al., 2010*).

In 2017, *Malik et al. (2017)* carried out an extensive study to analyze the structure and location of CALHM6 *via in-silico* tools. CALHM6 was revealed to be a membrane protein, with a single domain called Ca_hom_mod covering the amino acids ranging from

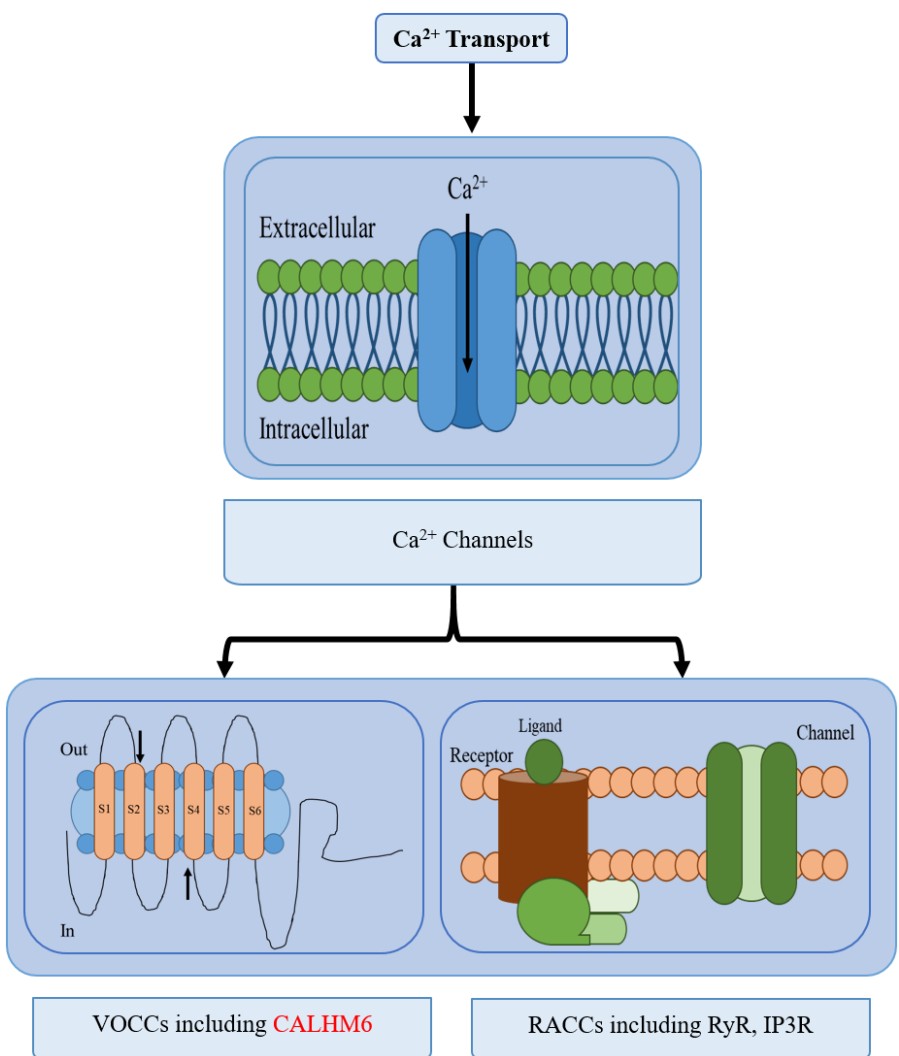

**Figure 1  Different types of calcium transport in a cell.** The different types of calcium transport in a cell include voltage-gated ion channels, receptor-activated ion channels, calcium pumps, and calcium exchangers. CALHM6 is included in voltage-gated calcium ion channels.

position 1–248 and found to be conserved during evolution, with the human CALHM6 to be most closely related to primates (*Malik et al., 2017*). Another study conducted by the group elaborated on the exact localization, expression, and all the proteins and molecules interacting with CALHM6 (*Malik et al., 2020*). It was reported to be localized in the Golgi-apparatus, with its non-conventional secretion to the plasma membrane in case of extracellular stress to produce the respiratory burst (*Malik et al., 2020*). CALHM6 was said to be the pore-forming unit of a voltage-gated ion channel, involved in calcium homeostasis during respiratory bursts and the production of reactive oxygen species through its calcium homeostasis modulator domain (*Malik et al., 2017*). A range of proteins was found to interact with CALHM6, including calcium binding proteins, cell adhesion molecules, transferases, nucleic acid-binding proteins, and cytoskeletal proteins (*Malik et al., 2017*).

**Table 1  Functions of genes of CALHM gene family.**

| Gene | Function | Source |
|---|---|---|
| CALHM1 | Role in detecting of Ca$^{2+}$ concentration, membrane voltage, Ca$^{2+}$ homeostasis, and cortical neuronal excitability. | Ma et al. (2016) |
| CALHM2 | Cation channel, play part in positive regulation of apoptosis, mediates ATP release in astrocytes, predicted to be part of the plasma membrane. | NCBI Gene Bank |
| CALHM3 | An important part of the ATP release channel in Type II taste bud cells (TBCs), and also mediates neurotransmitter release. | Ma et al. (2018, p. 3) |
| CALHM4 | Involved in channel activity of cations. | Syrjanen et al. (2020) |
| CALHM5 | Predicted to be involved in channel activity of cations. | Syrjanen et al. (2020) |
| CALHM6 | Transmembrane protein involved in calcium movement and regulation of various processes including apoptosis and immune cell signalling. | Malik et al. (2017) |

The most significant co-localization of CALHM6 was with calcium-binding proteins, a strong indicator of CALHM6's role in calcium homeostasis as a calcium channel. Through interaction studies (*Malik et al., 2020*) reported these five proteins to show the highest interaction with CALHM6: Calpain, Vinculin, protein S100-A7, Thioredoxin, Peroxiredoxin, and Calmodulin-like protein 5.

Domains have independent roles and evolutions that dictate the relationships of proteins through their interactions (*Vogel, Teichmann & Pereira-Leal, 2005*; *Zmasek & Godzik, 2011*; *Finn et al., 2014*; *Mistry et al., 2021*). A protein can contain only a single domain in isolation or can be a multi-domain protein (*Vogel, Teichmann & Pereira-Leal, 2005*; *Zmasek & Godzik, 2011*; *Finn et al., 2014*; *Mistry et al., 2021*). All of the protein domains identified so far have been put into large collections known as databases, which include SMART, CDD, COG, SCOP, TIGRFAM, and Pfam or public resources like Interpro that provide all the information regarding domain functions (*Finn et al., 2014*; *Gao et al., 2017*).

The information on domains is available on Pfam, a widely used database of protein domain families for sorting protein sequences into domains and families by employing probabilistic models generated from high-quality alignment sequences (*Albalat & Cañestro, 2016*; *Brito & Pinney, 2020*). The target sequence is searched against the database and sequence homology between sequences of the same domain family is identified following the reconstruction of their species-specific repertoires to reveal the pattern of their evolution. There are three main events in domain repertoire evolution: domain gains, duplications, and losses (*Gao et al., 2017*; *Brito, 2020*; *Eddy, 1992*; *Katoh & Standley, 2013*).

This is the first study designed to explore the evolution of protein domain repertoires of CALHM6. It aims to trace the specific events of gain, loss, and duplication in the CALHM6 protein domain, and the evolution of its interacting partners. A good knowledge of the evolutionary events of a protein and its domain family is thus of key importance in further elucidating its role in the immune system.

In this study, the evolutionary analysis of CALHM6 along with its other interacting partners as well as signaling molecules was conducted. For this purpose, all possible open

**Table 2  Details of organisms along with their accession numbers.**

| Sr. No. | Accession Ids | Scientific names | Common names |
|---|---|---|---|
| 1 | NW_013952945.1 | *Austrofundulus limnaeus* | Killifish |
| 2 | NW_005395513.1 | *Bos mutus* | Wild yak |
| 3 | NC_044518.1 | *Camelus dromedarius* | Camel |
| 4 | NC_051805.1 | *Canis lupus familiaris* | Dog |
| 5 | NW_004973439 | *Columba livia* | Rock dove |
| 6 | NC_007127.7 | *Danio rerio* | Zebra fish |
| 7 | NW_018388147 | *Exaiptasia diaphana* | Sea anemone |
| 8 | NW_004994914 | *Falco cherrug* | Saker falon |
| 9 | NC_018727.3 | *Felis catus* | Cat |
| 10 | NC_021674.1 | *Ficedula albicollis* | Collared flycatcher |
| 11 | NC_052534.1 | *Gallus gallus* | Red Jungle fowl |
| 12 | NC_000006.12 | *Homo sapiens* | Human |
| 13 | NW_017367071.1 | *Lates calcarifer* | Barramundi |
| 14 | NC_036790.1 | *Maylandia zebra* | Zebra Mbuna |
| 15 | NC_022034.1 | *Microtuso chrogaster* | Prairie vole |
| 16 | NW_018151314.1 | *Pogona vitticeps* | Central Bearded Dragon |
| 17 | NW_006533175.1 | *Python bivittatus* | Burmese python |
| 18 | NC_051355.1 | *Rattus Norvegicus* | Brown rat |
| 19 | NC_027301.1 | *Salmo salar* | Atlantic salmon |
| 20 | NW_019218192 | *Stylophora pistillata* | Hood coral |

reading frames were extracted from fully sequenced genomes of 20 organisms and a total of ~4,000 protein domains were determined. Then using a time-calibrated tree of organisms and domain matrix from a Python pipeline, the presence and absence of protein domains were reconstructed at the nodes of the tree by employing a parsimony model. This allowed us to analyze the evolution of the CALHM6 calcium modulatory domain and its interacting partners through domain gain, loss, and duplication events.

## MATERIALS AND METHODOLOGY

To identify CALHM6 encoded in different organisms, all genomic sequences for CALHM6 in each of the 20 organisms (Table 2) were retrieved from the National Center for Biotechnology Information (NCBI) gene databank.

A Python pipeline available as an open-source repository on GitHub was used to generate a protein domain matrix containing details of the presence or absence of domains in each genome (*Brito & Pinney, 2020*; *Brito, 2020*). This domain matrix was used for further phylogenetic analysis (Fig. 2).

### Protein domain identification using Python pipeline
#### *Retrieval of tORFs from Organism's Accession Ids'*

The pipeline was initiated with *tORFs.py i.e.,* the retrieval of all Open Reading Frames (ORFs) for each of the gene sequences of organisms. Its input was the list of accession ids from NCBI and names of the organisms in a text file (Table 2).
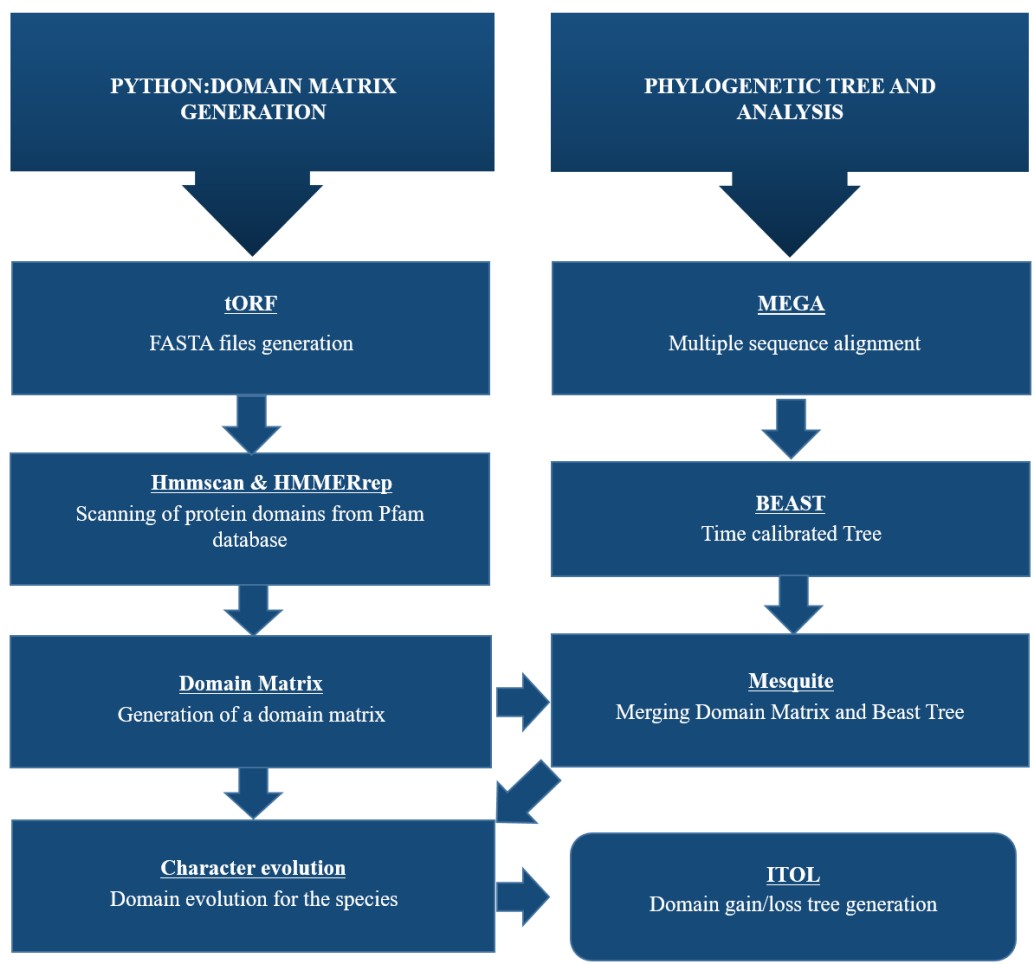

**Figure 2** **Schematic representation of methodology used for this study.** To identify CALHM6 encoded in different organisms, all genomic sequences for CALHM6 in each of the 20 organisms were retrieved from National Center for Biotechnology Information (NCBI) gene databank. A Python pipeline available as an open-source repository on GitHub was used to generate a protein domain matrix containing details of the presence or absence of domains in each genome. This domain matrix was used for further phylogenetic analysis and the final tree was visualized using iTOL.

### Retrieval of protein domains by scanning pfam database

The results proceeded to *hmmerrep2*. To run *hmmerrep2.py*, *hmmscan* was run first, which is a part of HMMER software (*Eddy, 1992*). Fasta files generated at tORFs were scanned for protein domain hits using *hmmscan* (*Mistry et al., 2021*). All Fasta sequences were scanned across HMM protein database *i.e.,* Pfam-A as a query sequence (*Finn et al., 2014*). To determine the reliable hits in scanned domains, inclusion criteria for both per-domain conditional value and per-sequence e-value was defined to be 0.001. (HMMER software is available as an online tool for short query sequences but for this study, it was accessed *via* supercomputer cluster, as the Windows version of HMMER was not available. It only runs on MAC or LINUX systems.)

### Retrieval of domain repertoires

By keeping the above-mentioned thresholds, the domain repertoires of each of the organisms were retrieved by giving the domtblout file as an input. *hmmerrep2.py* was run and the output was individual repertoire files for each organism (*Eddy, 1992*).

### Generation of domain matrix

*domMatrix.py* used the domain repertoires from previous results as an input, and a protein domain matrix was generated containing details of the presence or absence of domains in each genome. This matrix was used for further phylogenetic analysis.

## Phylogenetic analysis and protein substitution model prediction using ProtTest

To understand CALHM6 evolution across different species, the generation of a phylogenetic tree was performed. For this purpose, multiple sequence alignment of all protein sequences of all organisms was generated using MAFFT (*Katoh & Standley, 2013*). In the meantime, an amino acid substitution model for this alignment was generated using ProtTest, describing the probabilities of amino acid change, which selected an evolutionary model best suited for the reconstruction of protein phylogenies (*Eddy, 1992*; *Katoh & Standley, 2013*).

## Phylogenetic analysis

For Bayesian evolutionary analysis, the divergence dating function implemented in BEAST 2 software was used to generate a time-calibrated maximum clade credibility tree using a Markov Chain Monte Carlo Bayesian approach (*Jin & Brown, 2018*). Multiple alignments were imported as a partition for estimation of divergence time under Relaxed Clock Log Normal with JTT as amino acid substation model as estimated by ProtTest, and Yule speciation process as tree prior. Analysis was run for 6 million generations sampled every 10,000 generations. Tree annotator v2.6.6 with a burn-in percentage of 10% was used to generate a maximum clade credibility tree be proceeded further (*Eddy, 1992*; *Abascal, Zardoya & Posada, 2005*).

### Ancestral state reconstruction

The domain matrix generated in the Python pipeline and the time-calibrated tree from BEAST were further used in Mesquite to deduce the ancestral states (*Maddison, 2009*). The software used a linear parsimony reconstruction model to reconstruct the ancestral states. It treats domains as meristic (additive) characters. This model used the principle of maximum parsimony method to reconstruct the domains with a minimum number of evolutionary events, with the given domain distribution matrix and phylogenetic tree.

### Character evolution

The last procedure of our Python pipeline *i.e., charEvol.py* was the reconstruction of domain (character) evolution using the Mesquite matrix output (https://www.mesquiteproject.org/) along a given phylogeny. For this, protein domain counts per genome matrix were the character states. On this basis, mesquite estimated the domain gain, loss, and duplication on the tree branches employing the Parsimony model.

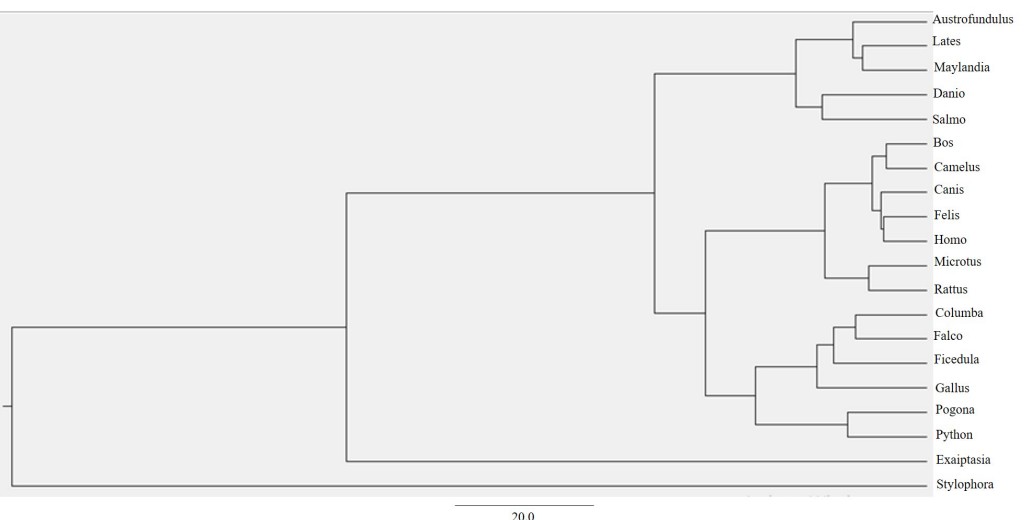

**Figure 3** **BEAST phylogenetic tree.** For Bayesian evolutionary analysis, the divergence dating function implemented in BEAST software was used to generate a time-calibrated maximum clade credibility tree using Markov Chain Monte Carlo Bayesian approach.

## RESULTS

### Protein domain identification using Python pipeline

*tORF.py* resulted in individual FASTA files for each organism whose accession id was provided, containing all protein sequences against each open reading frame.

*Hmmscan* found all the domain hits using HMMER and generated a text file named domtblout containing all the scanned domains using the Pfam database. Domtblout contained per domain data with each data line per homologous domain that was identified in a query sequence for each homologous model (*Eddy, 1992*).

Domain repertoires of each organism were retrieved from the given input domtblout file containing all the scanned domains. *Hmmerrep2.py* filtered out the results *i.e.,* unreliable hits in the domtblout file were excluded, and only those domains were exported which had domain E values and protein E values smaller than or equal to $1 \times 10^{-3}$.

The details of the presence or absence of domains in each genome were generated in the form of a protein domain matrix. This domain matrix was formed with the domain counts per taxa data.

### Phylogenetic analysis

Through ProtTest, the amino acid substitution model JTT+G+ was predicted which stands for the Jonas-Taylor-Thornton (JTT) model of protein evolution with a gamma distribution of rates (G+) (*Darriba et al., 2011*). This empirical model is widely used to estimate the amino acid substitution probabilities and evolutionary rates based on a set of protein sequences (*Darriba et al., 2011*). By using this model and multiple alignments from MAFFT, a maximum clade credibility tree using Markov Chain Monte Carlo Bayesian approach was generated (*Bouckaert et al., 2014*) (Fig. 3).

**Table 3  Details of domain gain, loss, and duplication events of CALHM6.**

| Nodes | Gain/loss/dup | Ca_hom_mod | Total events |
|---|---|---|---|
| n2 > n3 | Gain | 1 | 10 |
| n3 > n4 | Duplication | 3 | 97 |
| n14 > n15 | Duplication | 1 | 298 |
| n11 > salmo | Loss | 4 | 44 |
| n28 > n36 | Loss | 5 | 207 |
| n20 > n22 | Loss | 5 | 127 |
| n29 > n30 | Duplication | 2 | 15 |

## Ancestral state reconstruction and character evolution

The linear parsimony model reconstructs many domains at the internal node of the mesquite tree. This method helped in the detection of domain gain, loss, and duplication events of the protein domain, and the results were analyzed using a Python pipeline *i.e.,* character evolution. The output determined the domain evolution dynamics for each organism in the study. The output files generated contained the list of domain gain, loss, and duplication events reconstructed per branch of the tree. The events attained were visualized as a map along the branches of the tree using iTOL depicting the evolution of CALHM6 domain Ca_hom_mod repertoires (*Letunic & Bork, 2016*).

A total of 4,410 non-redundant sets of domains for all 20 organisms were retrieved as a result of character evolution and for these domains, around 14,000 domain gain, loss, and duplication events were recorded. The details of the events are mentioned below.

## Evolution of CALHM6 domain Ca_hom_mod

The evolutionary events centered on the Ca_hom_mod domain were screened from the events file generated through Python and given as input in iTOL to visualize the iTOL tree. The loss, gain, and duplication events recorded for CALHM6 are listed in Table 3. There were a total of 14 domain loss events, and six duplication events, while only one event was recorded where the domain had a gain. The duplication and gain of domain events increased progressively in the higher organisms, indicating that the increase in size and complexity of protein led to its functionality in higher mammals, especially humans.

In a rooted phylogenetic tree, the starting parent node corresponds to the common ancestor of all the species represented in the tree. The labels at the phylogeny's tips correspond to the specific organisms, while nodes are the branching points corresponding to the specific events. Each node represents the most recent ancestor of the lineages descended from that node.

In our study, the phylogenetic tree obtained from iTOL was also a rooted one. The main root was further divided into two branches. At one end of the branch, was the node of *Stylophora pistillata i.e.,* coral reef, the simplest of all organisms. There was no event of domain gain, loss, or duplication observed for *S. pistillata* indicating that it has the original CALHM6 from the common ancestor millions of years ago. For node 3 (n3), there was an event of domain gain on the node branch. n3 from the common ancestor was further

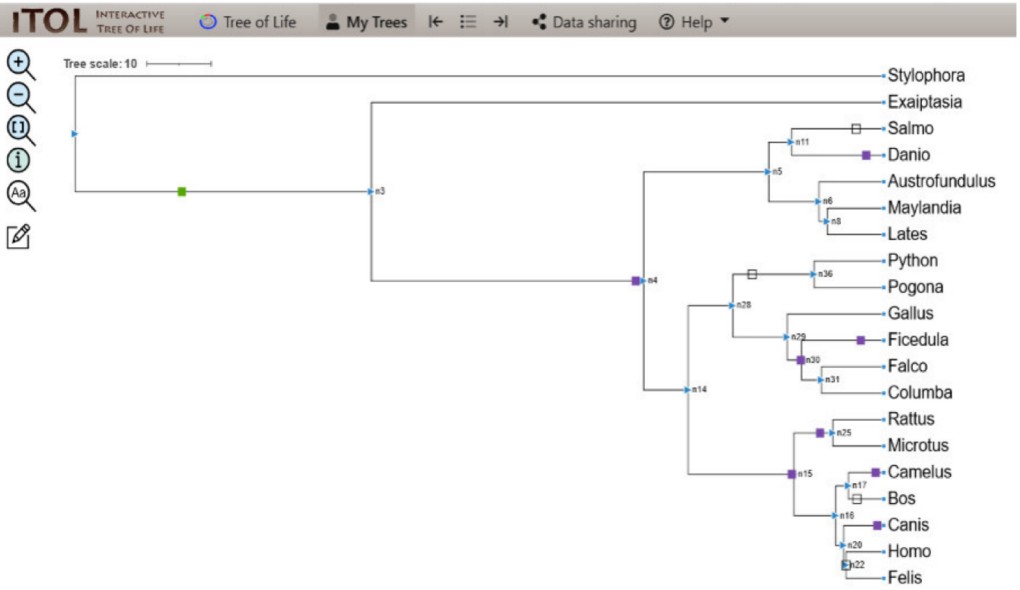

**Figure 4** **ITOL tree showing domain gain (green box), loss (white), and duplication (purple) events of CALHM6.** The evolutionary events centered on the Ca_hom_mod domain were screened. The loss, gain, and duplication events recorded for CALHM6 are shown here. There are a total of 14 domain loss events, and six duplication events, while only one event was recorded where the domain had a gain.

divided into two branches, *Exaiptasia diaphana* on one end and node 4 (n4) on the other end. There was no domain event detected for *E. diaphana* but at n4 there was a duplication event (shown as a purple box) occurring before dividing into n5 and n15. n5 contains 5 leaves at its end. *Lates calcarifer*, *Maylandia zebra*, and *Austrofundulus limnaeus* inherited the same CALHM6 domain as of n5 (duplicated one) but *Danio rerio* gained another duplication while *Salmo salar* observed domain loss (shown as a white box) of CALHM6 in it.

n14 is divided into two branches n28 and n15. n28 is further subdivided into n36 and n29. n28 observed domain loss before being divided into *Pogona vitticeps* and *Python bivittatus*. *Gallus gallus* inherited the same domain as n14. *Ficedula albicollis* observed two duplication events while *Falco* cherrug and *Columba* livia observed one duplication each.

n15 first observed domain duplication and was divided into n16 and n25. n25 is further divided into *Rattus Norvegicus* and *Microtus ochrogaster* but with another domain duplication.

n16 is divided into two clades. n17 containing *Camelus dromedarius* with domain duplication, and *Bos mutus* with domain loss. On the other clade, n20 is divided into *Canis lupis familiaris* with domain duplication while n22 contains *Felis catus* along with *Homo sapiens* with a domain loss event (Fig. 4).

## Domain evolution of interacting partners of CALHM6

Along with CALHM6, the Python pipeline yielded many protein domains. Through reported literature, major interacting protein partners of CALHM6 were shortlisted,

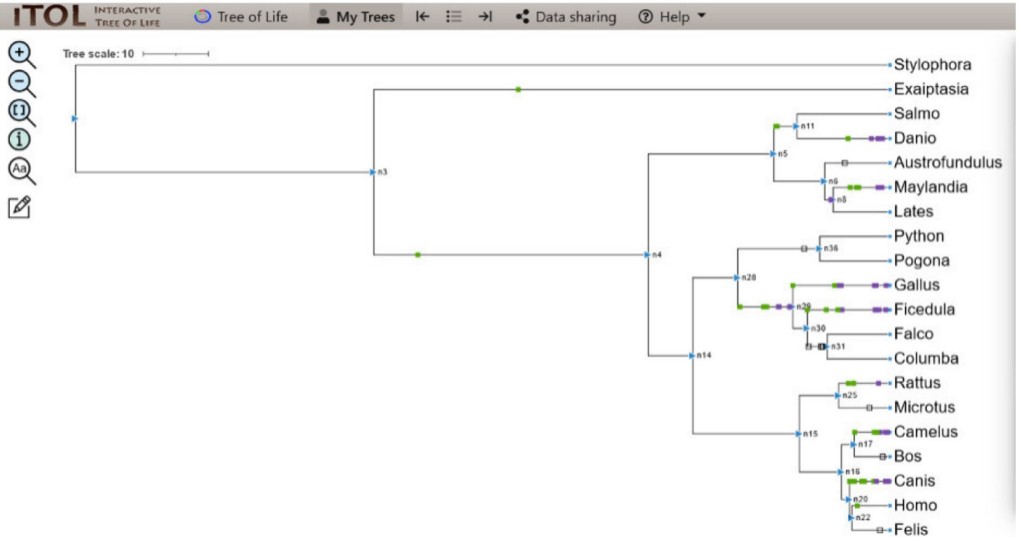

**Figure 5** **ITOL tree showing domain gain (green box), loss (white), and duplication (purple) events of CALHM6 along with its interacting partners.** The evolutionary events centered on the various interacting partners of the protein were screened. The loss, gain, and duplication events recorded for all of them are shown here.

which were majorly the calcium homeostasis proteins involved in the immunological phenomenon, and were searched for in the results (*Malik et al., 2020*). The presence of those proteins suggested a possibility of involvement of CALHM6 in calcium homeostasis and respiratory burst based on the similarity of specific motifs present in the protein.

The co-evolution of those proteins along with CALHM6 on the phylogenetic tree was determined. The partners included Calpain, Vinculin, protein S100-A7, Thioredoxin, Peroxiredoxin, and Calmodulin-like protein 5 (EF-Hand_1, EF-Hand_2, EF-Hand_3, EF-Hand_4 were selected as domains of Calmodulin-like protein 5). A total of 93 evolutionary events were recorded for these proteins. The iTOL tree is shown in Fig. 5.

For *S.pistillata*, there was no event of domain gain, loss, or duplication for any of the interacting partners. *E.diaphana* had an event of domain gain for Peroxiredoxin. EF-hand_1 had domain gain at n4. n4 divided into n5 and n14, with no events in both, and n5 further diverged into n6 and n11 where protein S100 observed a domain gain. For *S.salar*, there was no domain event, but for *D.rerio*, there was a gain, followed by duplication of Calpain-III, and duplications of S100, EF-hand_1, and Thioredoxin. n6, with no domain event, split into n8 with EF-hand_1 domain loss for *A.limnaeus*, and further its gain. n8 branched into *L.calcarifer* having no domain event, and *M.zebra* that had domain gain and duplication of S100, gains for Calpain-III and Thioredoxin, and duplication of EF-hand_1. n14 halving into n28, and n15 observed no event. n28 branched into n36 and n29 where there was a domain loss of EF-hand_1 at n36 but no domain event for *P.bivittatus* and *P.vitticeps*. There was domain gain and duplication at node 29 for EF-hand_4, gain for Calpain-III and Thioredoxin, and duplication of EF-hand_1.
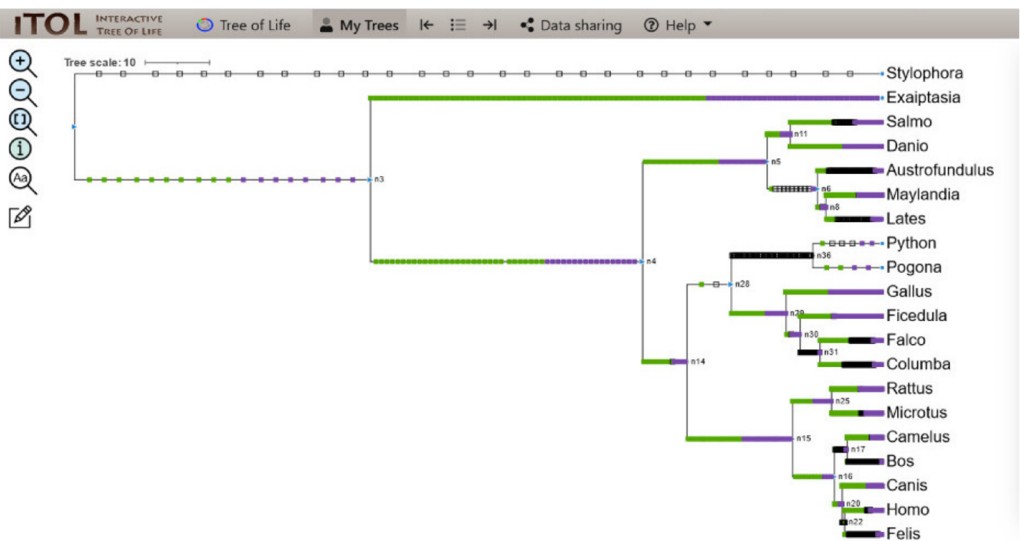

**Figure 6** ITOL tree showing domain gain (green box), loss (white), and duplication (purple) events of all the domains detected at the end of Python pipeline. The evolutionary events centered on all the domains were screened and the gain, loss, and duplication events recorded as a result are shown here.

Similarly, for *G.gallus*, EF-hand_2 gained domain with gain and duplications of EF-hand_3, following duplications of Calpain III, EF-hand_1, and EF-hand_4. n29 was split into n30 alongside *G.gallus* with subsequent branching into *F.albicollis* and n31. For *F.albicollis*, there was a gain of Vinculin, and duplications of EF-hand_1, Calpain III, and Thioredoxin, and both gain and duplication for EF-hand_2, and EF-hand_3. There were domain losses observed for EF-hand_1, EF-hand_4, Calpain III, and Thioredoxin at n31, but no events for *F. cherrug* and *C. livia*. n15 divided into n25 and n16 with no events observed. n25 forked into *R. norvegicus* that had domain gains of Peroxiredoxin, S100, and duplication of Vinculin. n16 dived into n17 and n20 with n17 further diving into *B. mutus* which had domain loss of EF-hand_1, and *C. dromedarius*, which observed domain gains of EF-hand_2, Thioredoxin, EF-hand_3. *R.norvegicus* had domain gain of Peroxiredoxin with no domain event for the *M.ochrogaster*. For *C. dromedarius*, there was domain gain of Thioredoxin. For *C.lupus*, there were gains and duplications of Calpain III, and Thioredoxin was observed. The only gain of Peroxiredoxin. Before the branching of n22 into *F.catus* and *H.sapiens*, there was no event detected for *F.catus*, and there was domain gain for Thioredoxin. Apart from these, all the domain events detected during the pipeline are also represented in Fig. 6.

## DISCUSSION

CALHM6 is an important immune protein that has recently gained attention as being a protein of immunological importance, with still room for exploration in various aspects. One such aspect was phylogenetic analysis across species, which has become an important tool throughout biology for comparing historical relationships and information about

genes, proteins, individuals, populations, and species. This information is usually in the form of morphological, behavioral, or molecular data, and depicted as a branching diagram, known as a phylogenetic tree. *Malik et al. (2017)* conducted an evolutionary analysis on CALHM6, showing it to be relatively conserved with the most homology with the chimpanzee. Additional *in silico* studies made by the same group revealed that CALHM6 was located on human chromosome 6, position 6q22.1, which translated to form a 315 amino acids long protein (*Malik et al., 2017*). However, no previous attempt to trace the exact domains and the evolution of CALHM6 had been made. The current study is the first study designed to explore the evolution of protein domain repertoires of CALHM6. It aims to trace the specific events of gain, loss, and duplication in the CALHM6 protein domain, concerning the evolution of its interacting partners across 20 different eukaryotic organisms.

The organisms chosen for analysis were made as diverse as possible to include all kinds of species and are shown in Table 2. The genomic ids were retrieved from NCBI and organisms with partial sequence submitted were excluded. It was admitted at the start of the Python pipeline which searched and gave the open reading frames as output which was presented as input for scanning the Pfam database. Different search algorithms like HMMER and RPS-BLAST are used to identify the required domain sequences from these domain databases. This study has used HMMER software for this purpose (*Eddy, 1992*). Likewise, superfamily and CDART algorithms can be used for domain architectures identification which is out of the scope of this article (*Fong et al., 2007*).

During the process of evolution, domain shuffling events may take place. These include domain gains, losses, and duplications. This shuffling helps in increasing the versatility of domain occurrence, functions, and protein complexity. Moreover, the concept of co-evolution of proteins involved in similar processes has been documented as an essential feature of evolution to maintain relationships of proteins with each other throughout evolution (*Holzerlandt et al., 2002*; *Brito, 2020*).

*Malik et al. (2020)* reported in 2021 that CALHM6 interacts with calcium homeostasis proteins, signaling molecules, *etc*. This study expanded on those findings and established an even stronger association between CALHM6 and its interacting partners through the possibility of the co-evolution of their domain repertoires. Since all possible open reading frames were annotated for all organisms, a result total of 4,000 domains were detected.

One of the possible reasons for ~4,000 protein domains detected by HMMER can be the homology between CALHM6 and the numerous other proteins in their sequence motif which have been involved in the same/similar pathways and interaction (*Mistry et al., 2013*). There is also a possibility of Ca_hom_mod domain similarity among various proteins and even the presence of the same domain in the proteins involved in similar pathways as CALHM6. The HMMER software package carried out deep multiple alignments of the input sequences and resulted in so many domains based on the similarity in the function and structure of CALHM6 with those proteins, revealing a specific pattern of evolutionary conservation. We took these results as an opportunity to visualize the evolution of the proteins previously reported in the literature as being an interacting partner of CALHM6.

In eukaryotes, domain shuffling events *i.e.,* gains, losses, and duplications are present in a proper equilibrium with each other but domain gains are found to be dominating at every branch of the tree of life. Keeping this in view, our study results have also shown domain gains and duplications mostly for interacting proteins but for CALHM6, there was a mixed trend involved. *Zmasek & Godzik (2011)* have also stated that eukaryotic complexity does not relate to the number of domains present. The reason mentioned is that complexity lies in the functional distribution of domains rather than just their number. *Zmasek & Godzik (2011)* found out that animal evolution at the genomic level is mostly related to protein domains associated with various processes including transcriptional regulation and several immune systems' functional aspects. About this, the interacting partners of CALHM6, mostly consisting of proteins involved in calcium homeostasis, that are essential in the immune system showed evolution patterns *via* domain gains, losses, and duplications (*Zmasek & Godzik, 2011*).

The domain matrix generated by the Python pipeline was incorporated with bioinformatics tools. CALHM6 protein sequences were obtained from NCBI and aligned using MAAFT and used to build a time-calibrated tree in BEAST. The tree was further used as input in Mesquite alongside the domain matrix, to conduct ancestral state reconstruction (ASR) and generate the final tree.

The ASR function in Mesquite was used due to various factors *e.g.*, vastness in genomic sequences data, availability of customized genes synthesis commercially, immenseness in bioinformatics tools for homologous sequence alignments, and phylogenetic determination of those sequences to describe their evolutionary relationships which made it feasible and user friendly (*Katoh & Standley, 2013*). Initially, the maximum parsimony (MP) method was usually used due to its easy execution for ASR. This method implies that the minimum substitutions in a phylogenetic tree should be the most likely. In recent ASR studies, probabilistic-based techniques like Bayesian reconstruction and maximum likelihood (ML) are also being used. The Bayesian approach has been reported to be more reliable than the maximum-likelihood method. Moreover, it provides the estimated confidence in the deduced ancestral state and specifies the posterior probability of amino acids (*Gumulya & Gillam, 2016*).

To address this uncertainty, the hierarchical Bayesian method estimates several ancestral state probabilities and averages them over all the possible evolutionary models and trees. For this study protein sequences were taken in ASR which are more reliable compared to nucleotide sequences. The sequences constitute diverse homologous sets from various evolutionary lineages to generate the phylogenetic relationship using a distance-based Bayesian approach by using empirical substitution evolutionary models (*Gumulya & Gillam, 2016*; *Del Amparo & Arenas, 2022*; *Holl et al., 2020*).

In this study, a total of 14,000 domain gain, loss, and duplication events were recorded. These domain events may contribute to either structural or functional attributes or even both. An interesting finding was seen with 14 domain loss events for CALHM6. According to *Zmasek & Godzik (2011)* domain losses are mostly associated with carbohydrates and amino acid biosynthesis processes, while Ca_hom_mod is involved in calcium homeostasis; this still opens up many possibilities for the CALHM6 role in the body. Six domain duplication

events are associated with increasing the number or amount of these domains in an organism's proteome, while only one domain gain event was recorded. The co-evolution with its calcium-interacting partners included Thioredoxin, Peroxiredoxin, protein S100-A7, Calpain, Vinculin, and Calmodulin. All of them combined yielded a total of 93 domain gain, loss, and duplication events which included mostly gain and duplication events.

The integration of Python and bioinformatic tools allowed for an effective approach to associate knowledge of domain contents and events of mammalian proteins and their phylogenetic history over a wide range of organisms. Although the generation of the domain matrix was time demanding, overall the approach was user-friendly and easy to execute. The ancestral character reconstruction was effective in providing valuable insight into exact domain events of CALHM6 alongside the evolution of its protein partners: Calpain, Vinculin, protein S100-A7, Thioredoxin, Peroxiredoxin, and Calmodulin-like protein 5. The pattern of their domain gain, loss, and duplications with perspective to domain events of CALHM6 gave a strong possibility of their co-evolution, further testifying the role of CALHM6 as a calcium homeostasis protein.

## CONCLUSION

CALHM6 is still a relatively new protein with much potential for exploration in various aspects. The current study applies the computational approach to trace the evolution of CALHM6s' domain involved in calcium homeostasis and compare the events with the proteins involved in calcium homeostasis and which have shown an association with it in previous studies. Evolutionary studies have been reported to be of importance, especially in deciding the drug target in various fields including molecular medicine and immunology. Commonly drug targets are relatively conserved during evolution, making them less prone to mutations and any possible change in site. This can lead to the future possibility of CALHM6 being used as a possible therapeutic target against diseases and conditions where it is found to be dysregulated and involved such as in various cancers and infections. Various inhibitors can also be designed against CALHM6 to improve the disease's prognosis and help recovery.

## ACKNOWLEDGEMENTS

We thank SINES, NUST for allowing us to use their supercomputing facility, Maheera Amjad for technical help with the command running on the supercomputer, and Faizan Habib, Aamir Ayub, Ayesha Fasih, and Dr. Ashar for providing help with the Python pipeline. We also like to thank Momina Ejaz for her helpful discussions and support. Last but not least, thanks to Anderson and John who made this work possible by making their Python codes publicly available.

### Funding

The authors received no funding for this work.

## Competing Interests

The authors declare there are no competing interests.

## Author Contributions

- Aneela Javed conceived and designed the experiments, analyzed the data, authored or reviewed drafts of the article, and approved the final draft.
- Sabahat Habib conceived and designed the experiments, performed the experiments, analyzed the data, prepared figures and/or tables, authored or reviewed drafts of the article, and approved the final draft.
- Aaima Ayub conceived and designed the experiments, performed the experiments, analyzed the data, prepared figures and/or tables, authored or reviewed drafts of the article, and approved the final draft.

## Data Availability

The sequences are available at NCBI: *Austrofundulus limnaeus*, NW_013952945.1; *Bos mutus*, NW_005395513.1; *Camelus dromedarius*, NC_044518.1; *Canis lupus familiaris*, NC_051805.1; *Columba livia*, NW_004973439; *Danio rerio*, NC_007127.7; *Exaiptasia diaphana*, NW_018388147; *Falco cherrug*, NW_004994914; *Felis catus*, NC_018727.3; *Ficedula albicollis*, NC_021674.1; *Gallus gallus*, NC_052534.1; *Homo sapiens*, NC_000006.12; *Lates calcarifer*, NW_017367071.1; *Maylandia zebra*, NC_036790.1; *Microtuso chrogaster*, NC_022034.1; *Pogona vitticeps*, NW_018151314.1; *Python bivittatus*, NW_006533175.1; *Rattus Norvegicus*, NC_051355.1; *Salmo salar*, NC_027301.1; *Stylophora pistillata*, NW_019218192.

The phyton codes are available at GitHub: https://github.com/andersonbrito/openData/tree/master/brito_2019_HVsDomains; https://academic.oup.com/ve/article/6/1/veaa001/5726995.

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
