# Peer review of "Evolution of protein domain repertoires of CALHM6"

_PeerJ, doi:10.7717/peerj.16063_

## Round 0.1 · original submission · Major Revisions

Please address the concerns of both reviewers and revise the manuscript accordingly.

Reviewer 1 ·

Basic reporting

Numerous grammatical errors and the complex/ compound sentence structure in this article make it exceedingly tough for readers to follow the contents. I highly recommend seeking assistance from a scientific English editing service to rectify the inconsistencies in the writing style and present the content in a polished and effective manner. The manuscript is currently weak in its present form. However, with extensive writing efforts, and meticulous interpretation of the data, there is potential for significant improvement.
The literature references provided are generally sufficient, and the figures and tables are presented in an acceptable manner. However, there is room for improvement in the figure legends/ captions to ensure a clear explanation of the figure contents.
The materials and Methods section is adequately described; however, it would benefit from some language refinement to improve clarity.

Experimental design

The manuscript demonstrates originality, accompanied by a well-defined problem statement and a clear focus on addressing a gap within the field of study. Although the presented data is sufficiently robust, discussion and explanation in several places are advisable for further enhancement.

Validity of the findings

I provided some corrections and a few queries to better understand the findings and to enhance the manuscript's quality.

Annotated reviews are not available for download in order to protect the identity of reviewers who chose to remain anonymous.
Cite this review as

Reviewer 2 ·

Basic reporting

The authors have reported the evolution of CALHM6 protein and its interacting partners across 20 different organisms using various bioinformatics techniques. Authors used multiple sequence alignments, python programming, accessing various computational programs such as iTOL, HMMER, MAFFT etc. to determine the evolutionary history and phylogenetic tree of CALHM6 and its interacting partners. The manuscript is accurate grammatically and well written. Tables and figures are in accurate format and are included where necessary.

Experimental design

Authors performed several computational analyses to obtain the evolutionary data on CALHM6 to ultimately build phylogenetic tree of its evolution and compare the events of gain, loss and duplication among 20 different organisms. The figure describing experimental pipeline is useful in understanding the flow and rationale of the experimental design.

Validity of the findings

The authors performed the evolutionary history analysis to understand the conserved domains in CALHM6 protein to use this data for potential therapeutic studies. The authors cited relevant studies and made use of available data and python code from GitHub to perform the experiments. Authors used inclusion criteria by applying e-value to validate the results during database scanning. For phylogenetic analysis, burn in % was used to provide high credibility to the data generated.

Additional comments

The manuscript is well written and can be accepted for publication. However, the organism names in the manuscript should be written in italics to follow the nomenclature standards.

Cite this review as

---

## Round 0.2 · accepted · Accept

The revised manuscript is acceptable now.

Reviewer 1 ·

Basic reporting

No comment

Experimental design

No comment

Validity of the findings

No comment

Additional comments

I am pleased to confirm that all my queries regarding the manuscript have been satisfactorily addressed. Based on the authors' revisions and the manuscript's contributions to the field, I recommend its acceptance for publication.

Cite this review as

Reviewer 2 ·

Basic reporting

The authors have reported the evolution of CALHM6 protein and its interacting partners across 20 different organisms using various bioinformatics techniques. The manuscript is accurate grammatically and well written. Tables and figures are in accurate format and are included where necessary.

Experimental design

The study analyzed the different domains of CALHM6 across 20 different organisms using extensive computational methods.

Validity of the findings

The research expands understanding of CALHM6 evolution and its interaction with calcium homeostasis proteins across diverse organisms. The manuscript discusses the importance of understanding the evolutionary history of CALHM6 to further exploit this target for therapeutics.

Cite this review as